# A game-theoretic analysis of networked system control for common-pool resource management using multi-agent reinforcement learning

**Arnu Pretorius**[*]
InstaDeep
Cape Town, South Africa

**Scott Cameron**[†]
University of Oxford
Oxford, UK

**Elan van Biljon**[†]
Stellenbosch University
Stellenbosch, South Africa

**Tom Makkink**[†]
University of Cape Town
Cape Town, South Africa

**Shahil Mawjee**[†]
University of Witwatersrand
Johannesburg, South Africa

**Jeremy du Plessis**
University of Cape Town
Cape Town, South Africa

**Jonathan Shock**
University of Cape Town
Cape Town, South Africa

**Alexandre Laterre**
InstaDeep
London, UK

**Karim Beguir**
InstaDeep
London, UK

## Abstract

Multi-agent reinforcement learning has recently shown great promise as an approach to networked system control. Arguably, one of the most difficult and important tasks for which large scale networked system control is applicable is common-pool resource management. Crucial common-pool resources include arable land, fresh water, wetlands, wildlife, fish stock, forests and the atmosphere, of which proper management is related to some of society's greatest challenges such as food security, inequality and climate change. Here we take inspiration from a recent research program investigating the game-theoretic incentives of humans in social dilemma situations such as the well-known *tragedy of the commons*. However, instead of focusing on biologically evolved human-like agents, our concern is rather to better understand the learning and operating behaviour of engineered networked systems comprising general-purpose reinforcement learning agents, subject only to nonbiological constraints such as memory, computation and communication bandwidth. Harnessing tools from empirical game-theoretic analysis, we analyse the differences in resulting solution concepts that stem from employing different information structures in the design of networked multi-agent systems. These information structures pertain to the type of information shared between agents as well as the employed communication protocol and network topology. Our analysis contributes new insights into the consequences associated with certain design choices and provides an additional dimension of comparison between systems beyond efficiency, robustness, scalability and mean control performance.[‡]

## 1   Introduction

Intelligent control systems based on multi-agent reinforcement learning (MARL) possess great potential for solving difficult tasks previously unmanageable by hand-engineered heuristic based

---

[*]Correspondence: `a.pretorius@instadeep.com`
[†]Work done during an internship at InstaDeep, Cape Town, South Africa.
[‡]The code for our analysis can be found at: https://github.com/instadeepai/EGTA-NMARL

systems. This is in part due to many recent MARL innovations utilising novel information structures, such as centralised training schemes to overcome issues of non-stationarity (Lowe et al., 2017) and learned networked communication protocols for scalable and effective cooperation (Foerster et al., 2016; Sukhbaatar et al., 2016; Chu et al., 2020). Moreover, critical infrastructure responsible for society's wellbeing have been demonstrated to be amenable to multi-agent system control and include management of electricity, telecommunications and transportation systems (Herrera et al., 2020; Haydari and Yilmaz, 2020; Chu et al., 2020).

A fundamental difference between traditional rule-based systems and those based on MARL is that the operating behavior of a MARL system is completely dependent on *learned* agent policies. These policies are in turn products of the process of agent interaction within an environment and depend on the specific information structures utilised by the designer of the system. In the context of networked MARL systems, we use the term *information structure* similar to Zhang et al. (2019), to refer to the type of information that gets shared between agents as well as the employed communication protocol and network topology.

**Game-theoretic analysis of networked system control**    Given that network MARL systems consist of learned agents, it becomes important that the designer of such a control system be able to understand the underlying incentives governing the interactions between agents and the operation of the system as a whole, especially in safety-critical control scenarios such as transportation or the management of life-supporting resources. To this end, we turn to game theory and consider using empirical game-theoretic analysis (EGTA) (Walsh et al., 2002; Wellman, 2006; Tuyls et al., 2018) as a viable approach towards better understanding networked MARL systems. Specifically, we analyse the learned operating behaviour of networked MARL systems for *common-pool resource* (CPR) management (Gardner et al., 1990) and investigate the following **research question (Q\*)**:

> **Q\***: *What are the game-theoretic solution concepts that arise from different information structures within a networked multi-agent reinforcement learning system used for common-pool resource management?*

**Common-pool resource management**    Motivated by the development in MARL technology and its applicability to system control, we focus on CPR as a particularly difficult but highly important task. CPRs are resources for which exclusion from access is difficult or impossible and where extraction of the resource by one agent diminishes what is available to be extracted by all remaining agents, at least for a certain time period (Ostrom, 1990). These resources are often renewable (regenerative) with either natural and/or artificial factors determining their rate of renewal but remain subject to complete exhaustion if appropriated at an unsustainable rate. CPRs constitute a large proportion of critical life-supporting resources including arable land, fresh water, forests, atmosphere and the climate.

In the case of self-interested agents, CPRs are subject to a particular social dilemma (Dawes, 1980) referred to as the *tragedy of the commons* (Hardin, 1968; Lloyd, 1833). Through recursive reasoning independent rational agents arrive at their best (Nash equilibrium) strategy which is to appropriate as much of the CPR as quickly as possible to the point of exhaustion (since all other agents are very likely to do the same). The tragedy then fully manifests when realising that had the agents instead cooperated towards a more sustainable use of the resource, every agent (even from a selfish perspective) would have been better off in the long run.

**Engineered networked systems as sequential social dilemmas**    Many recent works have focused on cooperation in the context of social dilemmas (Kleiman-Weiner et al., 2016; Peysakhovich and Lerer, 2017; Lerer and Peysakhovich, 2017; Peysakhovich and Lerer, 2018a,b; Foerster et al., 2018). This includes a fully-fledged research program investigating the emergence of cooperation in human-like MARL agents using EGTA (Leibo et al., 2017; Perolat et al., 2017; Hughes et al., 2018; Jaques et al., 2018; Köster et al., 2020). In particular, Leibo et al. (2017) use MARL and EGTA to extend the 2-player repeated matrix game social dilemma framework towards modeling *sequential social dilemmas* (SSDs). In an SSD, the strategic action to cooperate or defect is no longer atomic, but is instead associated with an intertemporally extracted strategy by an agent through policy learning. The strategic payoffs and corresponding equilibria for a modeled social dilemma situation involving two agents may then be estimated empirically as the expected return of a set of sampled policies, each representing either cooperation or defection. A similar analysis can be conducted for $N$-player SSDs using the approach of Schelling (1973), as was done by Perolat et al. (2017) and Hughes et al. (2018) to study the dynamics of MARL agents as humans interacting within a social community.

Here we consider SSDs purely from a networked systems engineering perspective. Our work is related to the study of games on networks (Jackson and Zenou, 2015), but is more focused on learning agents in the context of MARL. Specifically, we investigate the possibility of SSDs emerging from networked MARL control systems used for CPR management. Classical game theory literature has demonstrated that information structures such as direct information sharing and certain types of communication can alter the equilibria associated with different agent strategies (Roth and Malouf, 1979; Myerson, 1986; Farrell and Gibbons, 1989; Compte, 1998). Therefore, because an SSD is an environment and policy dependent phenomenon, we also expect different information structures in MARL systems for CPR management to have different game-theoretic solutions.

**Summary of our findings and contributions**     To the best of our knowledge, we conduct the first game-theoretic analysis of practical MARL systems for networked system control in the context of CPR management. Specifically, our analysis reveals the following

> **Answer to Q\***: *MARL systems display distinct equilibrium profiles that are dependent on their employed information structure. Systems with differentiable communication protocols tend to lead to improved agent cooperation, however, most system profiles still exhibit inefficiency at equilibrium. Interestingly, when using a neighbourhood weighted reward function, we find that the newly proposed NeurComm algorithm (Chu et al., 2020) is able to reach a stable equilibrium, where it is optimal for the system as a whole, as well as for each individual agent, to cooperate.*

Overall, our findings highlight the importance of networked control system design for effective CPR management. In particular, we reveal the interdependence between system design and the resulting game theoretic solution concepts that arise as a result of specific choices regarding the information structures employed within the system.

## 2   Methodology

Our investigation concerns the understanding of MARL for CPR management from a systems engineering point of view. Although inspired by previous work (Perolat et al., 2017; Hughes et al., 2018), which formed part of a larger game-theoretic driven research program on the emergence of cooperation in social communities of human-like agents. We instead study engineered networked systems comprising of general-purpose reinforcement learning agents, subject only to constraints such as memory, computation and communication bandwidth. However, similar to the above mentioned work, our agenda is solely in the realm of *descriptive science* as we attempt to answer Q\*. In other words, we do not provide any prescriptive innovations for networked MARL system control and leave this as a consideration for future work based on our findings. Rather, our goal in this work is to gain new insight and understanding of the operating conditions of learned networked MARL systems as well as the game-theoretic consequences related to their design.

**Networked $N$-player partially observable Markov games**     As modeling device, we consider partially observable Markov games where players are connected over a network. Our Markov game construction here is similar to the networked MDP defined in Chu et al. (2020). Specifically, we consider a graph $\mathcal{G}(\mathcal{V}, \mathcal{E})$ consisting of a set of nodes (vertices) $\mathcal{V}$ along with a set of edge connections $\mathcal{E} = \{(i,j)|i, j \in \mathcal{V}, i \neq j\}$, where each player is a node in the graph, locally connected to other player nodes. Players $i$ and $j$ from $\mathcal{V}$ are connected if the tuple $(i, j)$ is in the set $\mathcal{E}$. Each player has it's own local $d$-dimensional view of the global state $\mathcal{S}$ obtained through an observation function $O_i : \mathcal{S} \to \mathcal{O}_i \subseteq \mathbb{R}^d$, where $\mathcal{S} = \prod_{i=1}^{|\mathcal{V}|} \mathcal{O}_i$. Players take actions from their respective action sets $\mathcal{A}_i$, for $i = 1, ...|\mathcal{V}|$. To facilitate the use of communication along connections in the graph we define a communication space $\mathcal{C}$. Specifically, let the connected neighbourhood surrounding player $i$ be given by $\mathcal{N}_i = \{j \in \mathcal{V}|(i, j) \in \mathcal{E}\}$, then from the perspective of player $i$, the information communicated to it by its neighbours is given by the set $\mathcal{C}_i = \{m_{ji}|j \in \mathcal{N}_i\}$, where $m_{ji}$ represents a message being sent from player $j$ to player $i$. After taking an action, each agent receives a reward according to a neighbourhood weighted reward function $r_i : \mathcal{O}_i \times \mathcal{A}_i \times \mathcal{A}_{\mathcal{N}_i} \to \mathbb{R}$, with weighting parameter $\alpha$. Finally, let $p(\Delta)$ denote a probability distribution over a discrete set $\Delta$ and let the total number of players in the game be denoted by $N = |\mathcal{V}|$.

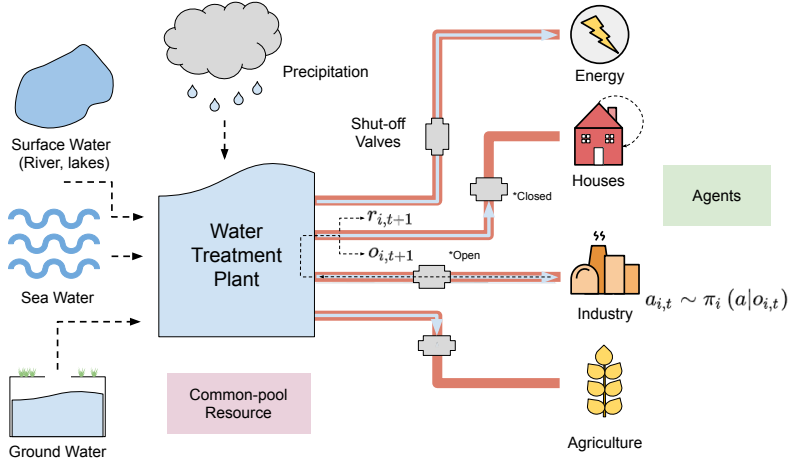

Figure 1: *Water management system as a common-pool resource environment for networked multi-agent reinforcement learning.* Shut-off valve controllers are treated as agents in the system and are responsible for providing water to different sectors of society, such as industry and agriculture.

We define a networked $N$-player partially observable Markov game $\mathcal{M}_\mathcal{G}$ as the following tuple $(\mathcal{G}, \{\mathcal{O}_i, \mathcal{A}_i, \mathcal{C}_i, r_i\}_{i \in \mathcal{V}}, \mathcal{T})$, where $\mathcal{T}$ is the global state transition function $\mathcal{T} : \mathcal{S} \times \mathcal{A} \rightarrow p(\mathcal{S})$ and $\mathcal{A} = \prod_{i=1}^{|\mathcal{V}|} \mathcal{A}_i$, represents the global action space. Player strategies rely on learned individual policies, $\pi_i : \mathcal{I}_i \rightarrow p(\mathcal{A}_i)$, where $p(\mathcal{A}_i)$ is a distribution over the action set $\mathcal{A}_i$ and $\mathcal{I}_i$ is an *information structure set* consisting of shared and/or communicated information, such as neighbourhood

$$\text{observations: } o_{\mathcal{V}_i,t} = \{o_{j,t}\}_{j \in \mathcal{V}_i}, \text{ where } \mathcal{V}_i = \mathcal{N}_i \cup \{i\},$$
$$\text{policies: } \pi_{\mathcal{N}_i,t} = \{\pi_{j,t}\}_{j \in \mathcal{N}_i} \text{ and/or,}$$
$$\text{messages: } m_{\mathcal{N}_i,t} = \{m_{ji,t}\}_{j \in \mathcal{N}_i}.$$

Players then take actions $a_{i,t} \sim \pi_i(a|I_{i,t})$, conditioned on the information structure set, $I_{i,t} \in \mathcal{I}_i$. Each player's goal is to maximise a connectivity weighted payoff computed as the following expected long-term discounted reward $\mathbb{E}\left[\sum_{t=1}^{T} \gamma^{t-1}(r_{i,t} + \sum_{j \in \mathcal{N}_i} \alpha r_{j,t})\right]$, where $\gamma$ is the chosen discount factor and $\alpha$ is a weight on neighbouring player rewards. For example, if $\alpha = 1$, connected players seek to maximise a shared global reward, whereas if $\alpha = 0$, players only care about maximising their own reward.

**Water management system as multi-agent CPR environment**    An example of a crucial life-supporting CPR is water. Institutional level management of surface and groundwater has shown to be particularly difficult, while at the same time growing in importance (Baudoin and Arenas, 2020). This is in large part due a growing need for systems to be more adaptable to external pressures from effects such as global climate change (Schlager and Heikkila, 2011). However, there is evidence to suggest that, at least at an institutional level, CPR management systems may benefit significantly from proper design (Sarker and Itoh, 2001; Sarker et al., 2009), based on well thought out design principles (Ostrom, 1990; Ostrom et al., 1991; Ostrom, 1993). Although these above mentioned studies concern many different stakeholders (far beyond a single engineered system), we envision that well designed networked control systems will play an ever more important role as a component of a larger institutional level intelligent CPR management system, not only for water, but for any type of CPR. Therefore, in this work, we focus solely on networked control systems for CPR management.

For our experiments, we designed a simplified model of a water management system, shown in Figure 1. Here the control components are treated as agents in the system and are responsible for providing water to different sectors of society, such as industry and agriculture, supplied along pipes connected to a water treatment plant. From the perspective of each agent controller, the water inside the treatment plant represents a CPR, i.e. controllers cannot be excluded from access by other controllers and the piping of water by one controller to a specific sector diminishes what is available to the rest. Agent actions include opening or closing a shut-off valve and reward is given for the amount of water an agent is able to supply to its respective sector. Specifically, opening the valve

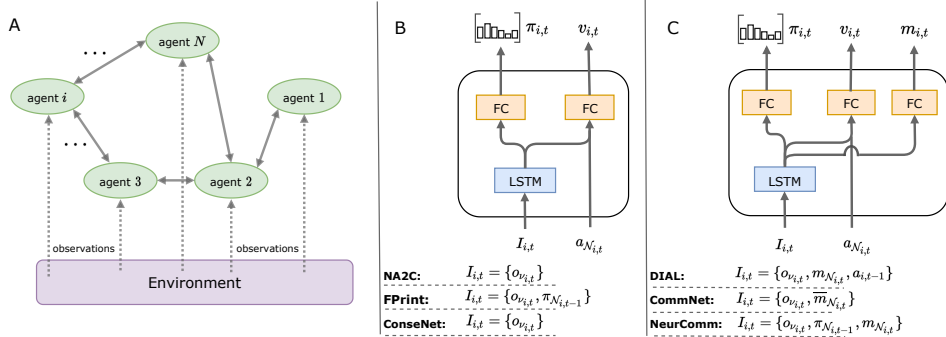

Figure 2: *Networked multi-agent reinforcement learning algorithms.* **(A)** Decentralised networked multi-agent system. Agents receive observations from the environment and share information with their neighbours. **(B)** Agent module for networked non-communicative algorithms (NA2C, FPrint, ConseNet). The agent module receives neighbourhood observation, action and/or policy information in order to compute a local policy and state-value estimates. **(C)** Agent module for networked communicative algorithms (DIAL, CommNet, NeurComm). The agent module has an additional fully-connected (FC) layer for message passing along differentiable communication channels.

allows an amount of $x_t = 0.1/N$ of water to flow through the pipe at each time step $t$, whereas closing the valve restricts all flow, i.e. $x_t = 0$, at each time step. At the beginning of each episode, the water plant starts off with an amount of water $w_0 = 0.5$ and regenerates at each time step at a constant rate of $w_{t+1} = w_t + c$ (we experiment with different values of $c$ in our experiments). The total number of agents in our system is $N = 4$ and each agent's observation space consists of a two-dimensional real-valued vector containing the value $w_t$ as well as the amount of water the agent has thus far piped to its sector. If the water is depleted, the episode ends. Each episode consists of a maximum of 1000 steps.

**Networked multi-agent reinforcement learning** The agents in our networked control system use reinforcement learning to learn how to map from states to actions. To study the effects of utilising different information structures within networked MARL systems, we consider six state-of-the-art approaches from recent work by Chu et al. (2020) who investigated the use of MARL for networked system control applied to adaptive traffic signal control as well as cooperative adaptive cruise control. All of the approaches make use of advantage actor-critic (A2C) with deep neural networks for function approximation (Sutton et al., 2000; Mnih et al., 2016). The first set of three algorithms are non-communicative wherein agents are restricted to observing only local information during execution and training, but do not explicitly communicate: **NA2C**, which is a networked A2C implementation of the MADDPG algorithm proposed by Lowe et al. (2017); **FPrint**, NA2C with policy fingerprints (Foerster et al., 2017); and **ConseNet**, NA2C with the consensus update proposed by (Zhang et al., 2018). The remaining three are communicative algorithms wherein agents both observe local information and communicate explicitly via differentiable communication channels: **DIAL** (Foerster et al., 2016); **CommNet** (Sukhbaatar et al., 2016); and **NeurComm** (Chu et al., 2020), where NeurComm was specifically inspired by the communication protocol proposed by Gilmer et al. (2017). Finally, we also consider fully independent learning using A2C, **IA2C**[2], where all agents are disconnected from each other and no information is shared, similar to IQL (Tan, 1993). All the algorithms are decentralised in the sense that none of them make use of a centralised critic or global policy network. However, the value estimates $\nu_{i,t}$ from the critic network for each algorithm are conditioned on shared neighbourhood actions $a_{\mathcal{N}_i,t} = \{a_{j,t}\}_{j \in \mathcal{N}_i}$. A summary depiction of the system design and network architectures as well as the type of information that gets shared between agents, for each of the above mentioned algorithms, is provided in Figure 2 (A-C).

A key motivation for a system designer to use MARL is to have the system learn its behaviour through reinforcement learning and potentially *self-discover* solutions significantly more efficient than those designed and implemented by a hand-engineered rule-based system. Moreover, beyond system

design and implementation considerations, an additional argument for using MARL is based on the following hypothesis. `Autocurriculum hypothesis` (Leibo et al., 2019): *The dynamics arising from competition and cooperation in multi-agent systems provide a naturally emergent curriculum, where the solution to one task often leads to new tasks, continually generating novel challenges, and thereby promoting innovation within the system.* There is evidence to suggest that this hypothesis could be true for reinforcement learning agents, e.g. in Baker et al. (2019), and therefore MARL could play a key role in sustained innovation for networked system control.

**Empirical game-theoretic analysis (EGTA)**    As multi-agent systems become increasingly reliant on learned complex behaviour, they will also become more difficult to analyse. Furthermore, these systems could be deployed for safety-critical operations or for managing crucial life-supporting resources, as in our example of water management. This makes understanding the underlying mechanisms driving the behaviour and interaction of the agents in these systems, all the more important. Here we make use of empirical game-theoretic analysis (EGTA) (Walsh et al., 2002; Wellman, 2006; Tuyls et al., 2018) as a tool for analysing networked MARL systems.

Game theory (von Neumann and Morgenstern, 1944) concerns the mathematical study of payoffs and incentives related to strategic actions taken by different agents within a particular setting represented as a game. However, a direct application of the theoretical tools from game theory for analysing a complex intertemporally learned multi-agent system can prove to be a very challenging task. In the context of general-sum Markov games, the one we consider here, classical tools of analysis in game theory are often more suited towards analysing strategic-form matrix games such as the *prisoner's dilemma*, i.e. games where agent strategies consist of atomic level actions that are directly labelled as either being cooperative or defecting, e.g. `confess` or `stay quiet`. Therefore, the networked control systems we consider in this work represent a significant challenge for classical analysis, however, we are able to overcome this challenge using more modern EGTA.

In EGTA, complex intertemporal strategic play in a multi-agent system becomes condensed into a *meta-level game*, where the atomic actions of the meta-level game correspond to learned agent policies. To achieve this, we follow the approach of Leibo et al. (2017), whereby we adjust the regeneration rate of our environment CPR so as to induce learned behaviour associated with either cooperation, or defection, and then estimate the average payoffs for different agents by evaluating these extracted strategies (see steps A-C in Figure 3). Specifically, we consider the level of *restraint* an agent is able to display under different regeneration rates as an indicator of cooperation or defection. We define restraint as the percentage of time spent with a closed shut-off valve. Under high regeneration rates, agents learn policies showing little restraint and continually extract from the resource, without any need to cooperate. We label these learned policies as defecting. In contrast, under low regeneration rates, it is possible for agents to learn policies that display higher levels of restraint by not extracting from the resource at certain times, which requires a larger degree of cooperation so as to not have the resource deplete. We label these learned policies, which display a high degree of restraint, as cooperative. Armed with these labelled policies, we consider a meta-level game, where binary choice analysis becomes possible. In particular, by having agents with different strategic incentives interact during play, we can make use of a meta-level analysis to find potential equilibria as well as strategic inefficiencies within the system.

**Intertemporally learned sequential social dilemmas (SSDs)**    Our aim is to identify possible networked system SSDs: operating conditions of a learned networked system under which at least one inefficient equilibrium exists among all the potential equilibria identified within a system. That is, there exists at least one instance where all agents could be made better off or the collective total payoff made larger by improved system organisation and cooperation. Similar to Perolat et al. (2017) and Hughes et al. (2018), we use EGTA combined with the binary choice analysis of Schelling (1973), to distill the complex interaction of a multi-agent system into a meta-level game from which the potential equilibria can easily be obtained. To achieve this, we specifically make use of *Schelling diagrams* (Schelling, 1973).

A Schelling diagram shows the average payoff for an individual agent choosing to cooperate, or defect, as a function of the total number of agents choosing to cooperate. An example of such a diagram is shown in step 4 of Figure 3, where the $x$-axis is in terms of the total number of agents choosing to cooperate. Concretely, if there are $N = 4$ agents and $x = 2$ are cooperating, this implies that $N - x = 2$ of the other agents are defecting. At this point, the Schelling diagram in Figure 3 will have three $y$ values: the average payoff (in a system with two cooperators and two defectors) for

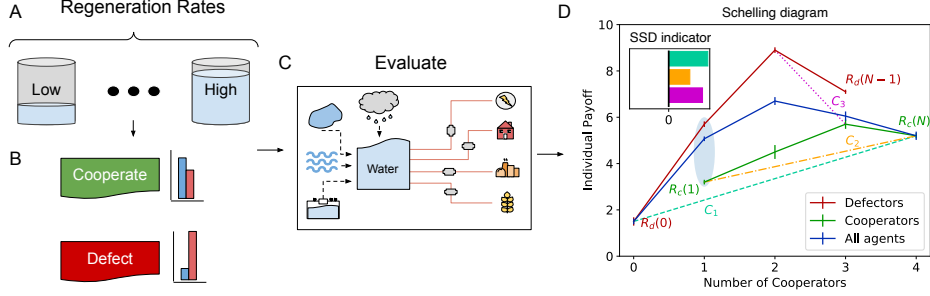

Figure 3: *Empirical game-theoretic analysis pipeline*. **(A)** Agents train on a common resource pool under different regeneration rates, where they may act greedily or show restraint. **(B)** Cooperative policies show restraint, while defecting policies act greedily. **(C)** Evaluate sampled policies on the environment. **(D)** Plot the Schelling diagram, which can be used to visually classify systems.

an agent choosing to either (1) cooperate (green line denoted $R_c(x)$) or (2) defect (red line denoted $R_d(x)$) as well as (3) the average (blue line) for all agents (cooperators and defectors). To identify an SSD, we use similar criteria as in (Macy and Flache, 2002; Leibo et al., 2017; Hughes et al., 2018), but collapse the specific conditions referred to as "fear" and "greed" in prior work, into a single criterion.[3] In doing so, we make use of the following three conditions which need to be satisfied. *Condition 1*: full system cooperation is preferred to full system defection; *Condition 2*: full system cooperation is preferred to being exploited by defectors; and *Condition 3*: for a certain range of system configurations there is a stronger reward-driven incentive to defect than to cooperate. These conditions are summarised as follows:

$$C_1 : R_c(N) - R_d(0) > 0$$
$$C_2 : R_c(N) - R_c(0) > 0$$
$$C_3 : R_d(n-1) - R_c(n) > 0,$$

for all $n \leq i$ and/or $n \geq j$ for some $i, j \in \{1, ..., N\}$ with $j \geq i$. If all of $C_1$ to $C_3$ are satisfied we classify the system operating conditions as an SSD. The inset in our Schelling diagram shown in Figure 3, which we refer to as an *SSD indicator*, computes the numerical value for each of the above conditions and immediately indicates the presence of an SSD if all the values are positive (i.e. all three horizontal bars are pointing to the right). To compute the value in the SSD indicator related to $C_3$, we use the value that is the maximum of $R_d(n-1) - R_c(n)$ over the range specified above. Finally, we use shaded ovals to mark the payoffs associated with the potential equilibria in a system.

## 3 Results

The first stage of our analysis was concerned with extracting policies learned under different CPR regeneration rates, $c = \{0.1, 0.088, 0.077, 0.065, 0.053, 0.042, 0.03\}$, using our water management environment. The values in the heatmap shown in Figure 4 are the average percentage restraint displayed by the tested MARL algorithms for different regeneration rates, with a neighbourhood reward weighting set at $\alpha = 0.1$ (we consider $\alpha = \{0, 1\}$ in the supplementary material). From a system design perspective, a value of $\alpha = 0.1$ ensures that agents remain focused on providing water to their respective sectors, while at the same time taking into consideration the water needs of their connected neighbourhood. As expected, agents show less restraint at higher regeneration rates. However, at lower regeneration rates, differences start to emerge. This can be seen mostly between the algorithms that employ differentiable communication protocols and those that do not. Specifically, communicating algorithms seem to show less restraint at lower regeneration rates compared to other approaches, which could possibly be attributed to more sophisticated cooperation strategies. In contrast, for algorithms that do not communicate, but only directly share local information, it might prove more difficult to coordinate agent behaviour in the low resource setting and as a result each agent is forced to show a higher level of restraint to keep the resource from the depleting. That said,

we also observe a marked decrease in restraint between the rate = 0.042 (second to last column) and the rate = 0.03 (last column) for three of the non-communicating algorithms (IA2C, NA2C and FPrint). This shows that at an extreme level of scarcity, these agents find it difficult to learn restraint, leading to myopic behaviour and the tragedy of the commons to manifest.

In the second stage of our analysis, we classified policies into two sets: cooperating or defecting, based on their level of restraint. We then estimated the expected payoffs related to each policy combination in our four-agent water management system, with a constant regeneration rate of $c = 0.055$ and 100 steps per episode. At this rate of regeneration, we labelled policies as defecting if they had a percentage of restraint below $25\%$ and cooperating if this percentage was instead above $35\%$. The results of our analysis are presented in Figure 5 (A-G) in the form of Schelling diagrams, one for each algorithm, from which the game theoretic solution concepts can be visually obtained. Our parameterisation for these diagrams are in terms of the potential payoffs (shown on the $y$-axis) for agents (cooperating or defecting) with respect to the *total* number of cooperators (on the $x$-axis) as in Perolat et al. (2017), however, we also provide the alternative parameterisation using the number of *other* cooperators on the $x$-axis, as in Hughes et al. (2018), in the supplementary material.

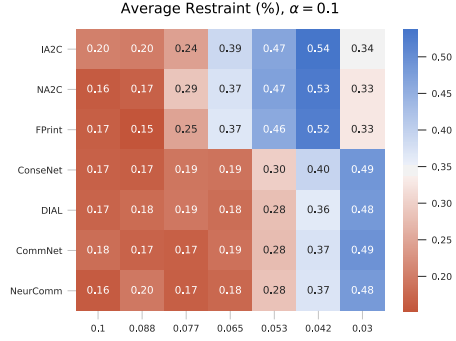

Figure 4: *EGTA for networked system control, with $\alpha = 0.1$.* Heatmap of average restraint percentage as a function of the regeneration rate for different MARL algorithms, from high $(0.1)$ to low $(0.03)$.

The shaded blue ovals in the Schelling diagrams of Figure 5 mark the various payoffs for cooperating and defecting agents, associated with potential Nash equilibria for each system. For example, in IA2C shown in panel (A), there exist two Nash equilibrium points. These are best response strategies from the perspective of each individual agent, given the strategies of all other agents. At the first equilibrium point, the system consists of two cooperating and two defecting agents. In this situation, a cooperating agent will not receive a higher payoff for switching to defect, and neither will a defecting agent by switching to cooperate. However, note that the global expected payoff is not optimal in this case and had all agents instead started out by cooperating (represented by the second equilibrium point with payoffs shown at the top right of the plot), agents would have no incentive to switch and at the same time achieve the global maximum expected payoff for the system as a whole. That said, both these equilibrium points are unstable. For instance, if the system was to be perturbed towards a configuration of three cooperators and one defector, it is unclear whether the system would return to the same equilibrium point.

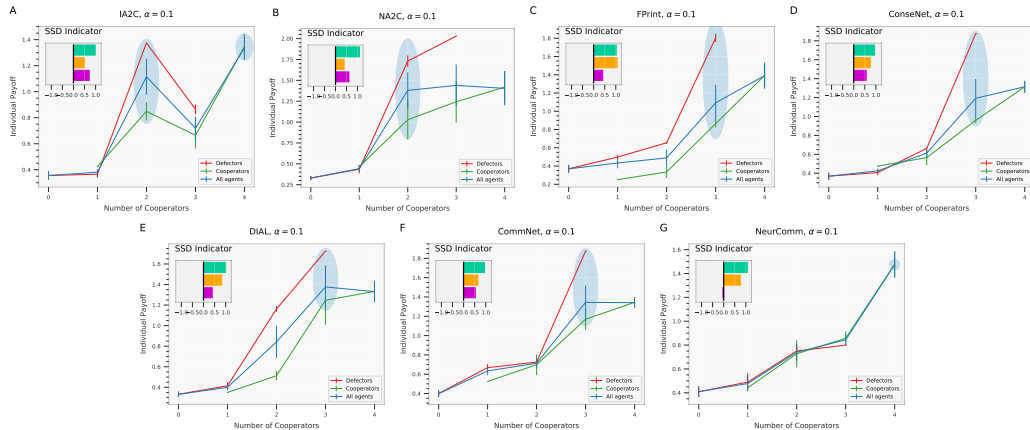

Figure 5: *EGTA for networked system control, with $\alpha = 0.1$.* **(A-G)** Schelling diagrams for each approach with sequential social dilemma (SSD) indicators given as insets. Potential equilibria are shaded in blue.

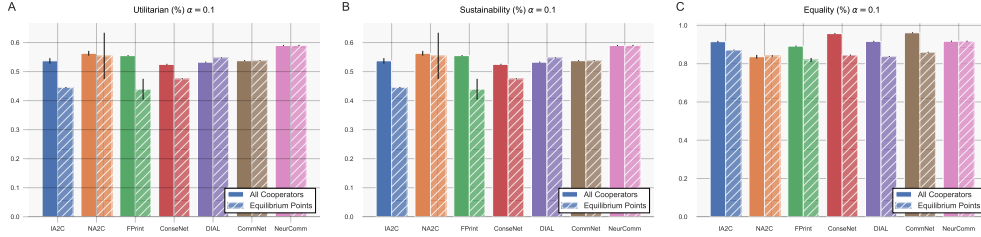

Figure 6: *Social metrics at equilibrium.* **(A)** Utilitarian **(B)** Equality **(C)** Sustainability

Although Nash equilibria are useful for investigating optimal best response strategies from an individual agent perspective, with $\alpha = 0.1$, the reward for each agent in our case depends on the rewards of its entire neighbourhood. Therefore, our interest mostly concerns the expected payoff of the system, represented by the average payoff for all agents (the blue lines in Figure 5). Even though many of the equilibria in Figure 5 are inefficient from this perspective, we find that for communicating algorithms their equilibria do in fact coincide with what is optimal for the system as a whole (see B,E and F). However, in the case of DIAL (E) and CommNet (F) we still consider these equilibria to be inefficient as indicated by our SSD conditions. This is because if we make the assumption that all water demanding sectors of society are considered as being equally important, these equilibria still represent an unequal distribution of the resource (the difference between cooperators and defectors). Interestingly, NeurComm (G) is the only non-SSD among all the algorithms tested ($C_3 < 0$, i.e. there exist no reward-driven incentives for agents to defect). That is, NeurComm is the only algorithm able to achieve a stable equilibrium point, where what is optimal for each individual agent is also optimal for the entire system.

In Figure 6, we compute the metrics from Perolat et al. (2017), namely, the Utilitarian, Equality and Sustainability metric for each algorithm for the two cases where (1) all agents cooperate and (2) agents play their equilibrium strategy. When all agents are cooperating, the equality for NeurComm is lower (i.e. the distribution of reward among agents is less uniform) than that of the other communicating algorithms, however, it is still able to achieve both a higher score for the group (more utilitarian) and have higher levels of sustainability. This provides some supporting evidence that NeurComm is perhaps better able to learn how to coordinate effectively.

## 4 Discussion

Networked multi-agent reinforcement learning has shown to be a viable option for tackling large-scale control problems. However, due to the complexity of learned multi-agent systems, gaining insight into their operating behaviour can prove to be difficult. Moreover, these systems are likely to be increasingly deployed in a wide range of important settings, making their understanding ever more important. In this work, we considered the setting of common-pool resource management. Control problems of this type relate to some of society's greatest challenges including food security, inequality and climate change.

We conducted an empirical game-theoretic analysis of networked multi-agent systems for common-pool resource management. This analysis highlighted the differences in solution concepts that arise from different information structures used in these systems. Specifically, we found that differentiable communication protocols play an important role in driving the system dynamics to equilibrium points associated with optimal payoffs, both for the individual and the system as a whole. However, in terms of maintaining an equitable distribution of resources amongst agents, most of these equilibria were still regarded as being inefficient. In the end, NeurComm (Chu et al., 2020), a newly proposed networked multi-agent reinforcement learning algorithm, was the only approach able to obtain an optimal equilibrium that was both stable and efficient.

Finally, in addition to highlighting the interaction between system design and behaviour, we consider this work as demonstrating a viable evaluation pipeline for complex multi-agent systems beyond efficiency, robustness, scalability and mean control performance.

## Broader Impact

It could be extremely costly should a critical multi-agent system completely fail. However, in this work, we have shown that even in a very simplified environment, a seemingly working system might convergence towards equilibriums that are still inefficient or unequal. These outcomes are also clearly undesirable, yet they do not provide a clear signal for system failure. Given the complexity of MARL system operation, failure modes related to inefficiencies at equilibrium are more subtle, and more difficult to detect.

We consider our work a contribution towards better understanding networked multi-agent reinforcement learning systems for common-pool resource management. As mentioned in the main text, we envision these systems becoming more widespread in their use as effective control systems for managing critical life-supporting resources. However, a specific challenge facing future systems is to have them be highly adaptable. Multi-agent reinforcement learning offers this capability, but it also makes systems far more difficult to analyse and understand. Therefore, even though we consider our environment a considerable simplification over a real institutional level CPR management system, we nevertheless hope the broader impact of our work is to demonstrate EGTA as a viable approach to analysing practical multi-agent reinforcement learning systems.

## Funding Disclosure

KB is the CEO of InstaDeep. AP and AL are employees of InstaDeep. SC, EB, TM and SM were funded by InstaDeep during an internship. In addition, TM is the recipient of financial support through the Ada & Bertie Award, Mullne, Rose & Sydney Award and a University of Cape Town Vice Chancellor's Research Scholarship.

## Footnotes

[2]In Chu et al. (2020), NA2C is referred to as IA2C and our IA2C (fully independent/disconnected learning) was not considered in that work.

[3]In the context of an engineering system, these two conditions would seem an anthropomorphisation of the incentive to defect.

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
