[Supplementary Material]

## Supplementary Material

We provide additional results for EGTA applied to networked MARL system control for CPR management. Specifically, we investigate the consequence of different reward structures. As mentioned in the main text, each player in our $N$-player networked Markov game seeks to maximise a connectivity weighted payoff computed as the following expected long-term discounted reward $\mathbb{E}\left[\sum_{t=1}^{T} \gamma^{t-1}(r_{i,t} + \sum_{j \in \mathcal{N}_i} \alpha r_{j,t})\right]$, where $\gamma$ is the chosen discount factor and $\alpha$ is a weight on neighbouring player rewards. If $\alpha = 1$, connected players seek to maximise a shared global reward, whereas if $\alpha = 0$, players only maximise their own reward. Finally, for $0 < \alpha < 1$, players take into consideration a proportion of the rewards obtained by their connected neighbourhood while also maximising their own reward. Here we show results for $\alpha \in \{0, 0.1, 1\}$.

**Restraint percentages under different regeneration rates** The heatmaps in Figure 7 (A-C) highlight the differences in restraint percentage for different values of $\alpha$ as the regeneration rate is changed from high $(0.1)$ to low $(0.03)$. In the case where agents are completely self-interested $(\alpha = 0)$ shown in (A), the majority of algorithms without communication display very low levels of restraint for all rates of regeneration. To some degree, this could be seen as a manifestation of the tragedy of the commons where the equilibrium strategy for self-interested agents is to have zero restraint, especially when the regeneration rate is low, i.e. 0.042 or 0.03. In contrast, when connected agents completely share their reward $(\alpha = 1)$, shown in (C), all algorithms display lower levels of restraint when the regeneration rate is high, and higher levels of restraint when the regeneration rate is low. However, there is still some difference in the level of restraint between agents that communicate and those that do not, with the former showing slightly less restraint for lower levels of regeneration than the latter. This could possibly be attributed to better coordination as well as cooperation between agents that communicate, making them able to extract more of the resource closer to the limit of its capacity without depleting the resource completely.

Figure 7: Heatmaps of average restraint percentage as a function of the regeneration rate for different MARL algorithms, from high $(0.1)$ to low $(0.03)$. **(A)** $\alpha = 0$, **(B)** $\alpha = 0.1$, **(C)** $\alpha = 1$.

Figure 8: Schelling diagrams for each approach with network system sequential social dilemma (SSD) indicators given as insets. Potential Nash equilibria are shaded in blue. **Top row (A-G)** $\alpha = 0$, **Middle row (H-N)** $\alpha = 0.1$, **Bottom row (O-U)** $\alpha = 1$. Here we include orange shaded regions indicating configurations corresponding to the highest average payoff for all connected agents.

**Schelling binary choice analysis for different $\alpha$ values**   We performed the $N$-player binary choice analysis of Schelling (1973) for the different reward structures corresponding to $\alpha \in \{0, 0.1, 1\}$ and plot the Schelling diagrams for each value of $\alpha$ in Figure 8. The diagrams in the top row, panels (A-G), are for the different algorithms with $\alpha = 0$ (self-interested agents), the middle row (H-N) with $\alpha = 0.1$ (as in the main text) and the bottom row (O-U) with $\alpha = 1$ (global shared reward for connected agents). For self-interested agents ($\alpha = 0$) without communication, the insensitivity to the regeneration rate can cause the restraint threshold for classifying agents as cooperative or defective, to never be low enough to obtain all possible configurations of agents. This can be seen in the top row, panels (A-C), where for IA2C, NA2C and FPrint there were no instances where all agents could be considered as being cooperative. However, even for the communicating algorithms where cooperation is seen to emerge more easily, the majority of potential equilibria are still inefficient. In fact, only DIAL and CommNet have potential equilibria points that correspond to full system cooperation with expected payoffs that are optimal for the individual as well as the group. Also worth noting is that the equilibrium profile for ConseNet is similar to the communicating algorithms, DIAL, CommNet and NeurComm (across all values of $\alpha$), which is likely due to its consensus update mechanism.

The Schelling diagrams for the different algorithms with connected agents sharing a global reward ($\alpha = 1$) are shown in the bottom row of Figure 8. The orange ovals in these diagrams indicate which system configurations correspond to the highest expected payoff for all agents. In the case of non-communicative algorithms, IA2C, NA2C and FPrint, agents received on average the highest payoff when the system consisted of a mixture of cooperative, as well as defective, agents. In contrast, ConseNet and the communicating algorithms, DIAL, CommNet and NeurComm, had their highest payoffs coincide with systems operating at full cooperation.

**Schelling diagrams using a different parameterisation**   An alternative parameterisation for a Schelling diagram is to plot payoffs for a particular agent (cooperating or defecting) with respect to the number of *other* cooperators on the $x$-axis, instead of the *total* number of cooperators. We find the latter (which we use in the main text) more suitable for highlighting payoffs associated with potential equilibria, but note that the former provides an easier visual interpretation of dominant strategies for any given situation. We provide this version of the diagram for each algorithm for the case of $\alpha = 0.1$ in Figure 9.

Figure 9: *EGTA for networked system control ($\alpha = 0.1$) with the number of other cooperators shown on the x-axis.***(A-G)** Schelling diagrams for each approach with sequential social dilemma (SSD) indicators given as insets.

A trend easily observed using this parameterisation is that for most algorithms the dominant strategy for a learned agent is to cooperate until all agents are cooperating, where at this point, the dominant strategy switches to defect. The only exception is the NeurComm algorithm, which is shown that have cooperation as the dominant strategy for any configuration of the system.

**Finite sample analysis using bootstrap estimation**   It is possible to connect our analysis to the underlying Markov game. More specifically, a key result for EGTA is given by the finite sample analysis in Tuyls et al. (2018), which states that given enough samples it is possible to bound the difference between the empirical equilibrium payoff estimates and those obtained in the original

game. We used this result combined with bootstrap resampling to more tightly bound our estimation difference and used these improved estimates in our presented results. Figures 10 to 16 show histograms of the bootstrap procedure with mean payoff estimates for the different algorithms for the case of $\alpha = 0.1$. We note that not all estimates could be improved. In a few cases the original samples we obtained displayed very low variance, sometimes even with an effective sample size of only one and zero variance. Therefore, in these low sample diversity cases, our estimates are less reliable as an improvement on the original payoff estimate. For example, when the number of other cooperators is zero in IA2C (Figure 10), the resampling distribution over payoffs when defecting approaches a normal distribution, whereas for cooperation, it is a constant with zero variance.

Figure 10: **IA2C**

Figure 11: **NA2C**

Figure 12: **FPrint**

Figure 13: **ConseNet**

Figure 14: **DIAL**

Figure 15: **CommNet**

Figure 16: **NeurComm**