[Reviews · NeurIPS 2020]

Review 1

Summary and Contributions: Presents an environment and methodology for describing the learning dynamics of agents in network game social dilemmas. Evaluates the performance of a number of communication algorithms in these settings, demonstrating that these converge to qualitatively and quantitatively different equilibria.

Strengths: 1. Extends a popular line of recent literature on sequential social dilemmas to the setting of networked communication. 2. Evaluates performance of a wide range of relevant algorithms, providing a good coverage of recent methods. 3. Mainly well-explained and well-motivated with a clear and reproducible evaluation methodology.

Weaknesses: 1. Insufficient analysis of the results, for example the claim on line 274 should be further substantiated by looking at the data more closely, or by conducting further experiments. 2. Unclear as to the definition of NSI. Why is this obvious equivalent to the social dilemma conditions? It is not clear what the introduction of the term NSI is adding here. 3. There are some missing baselines, most obviously the RIAL algorithm. 4. There are some minor mistakes, for instance, the equilbrium in figure 4D seems to be incorrectly identified. 5. The literature on learning in network games could be more extensively cited. For example this paper seems relevant: https://perso.amse-aixmarseille.fr/bervoets/publis/WP_BervoetsBravoFaure.pdf.

Correctness: Yes.

Clarity: Yes.

Relation to Prior Work: Yes.

Reproducibility: Yes

Additional Feedback: This is a good paper, but more detailed analysis, and some tightening of the text is required to make it a strong candidate for acceptance. Suggestions for improvement are found in the "weaknesses" section. Response to authors: many thanks for responding to my points. The additional analyses you provide are promising, but I feel that they are more "descriptive" than "causal". In future versions of the paper, you should consider providing and testing specific hypotheses about why particular types of communication alter the incentive structure in such a way that the equilibria change. In response to your introduction of the term "NSI", I would caution against adding a new term for an existing concept, even when you find an interesting new interpretation of the concept. An alternative that would help unify the field would be to discuss this paper as a practical illustration of the SSD concept, and indicate how SSDs may be profitably interpreted as modelling networked system inefficiencies. I remain of the opinion that this work is above the submission threshold. Whatever the outcome of the review process, I would encourage the authors to keep pursuing this interesting line of work, particularly if they are able to scale the algorithm up to further environments.


Review 2

Summary and Contributions: This paper studies the networked multi-agent reinforcement learning systems for common-pool resource management. The authors experiment on the water management environment with six MARL algorithms. Furthermore, they conduct an empirical game-theoretic analysis of networked multiagent systems.

Strengths: This paper gives some insightful experimental analysis.

Weaknesses: However, there exist some weaknesses in this paper. W1: There is only one scenario provided in this paper (i.e., water management environment). It would be better if more environments such as the commons game “apples” in [1] could be investigated. W2: For the common-pool resource management and networked multiagent system, the 4-agent setting may be too small. W3: Why does the NuerComm achieve a stable equilibrium point while other algorithms cannot? Is there any explanation or analysis for it? W4: Some social outcome metrics are proposed in the previous work [1] such as Utilitarian and Equality. Why not use these metrics to analyze the performance of MARL algorithms in the common-pool resource management problem? Could you compare these metrics with yours to show some advantages? W5: The non-communicative algorithms perform better than the communicative algorithms with the regeneration rate at 0.042. But it is strange to see the opposite situation with the regeneration rate at 0.03 while both regeneration rates are low. It is better if you could provide more explanations about it. W6: Figure 4 is too small and some typos exist such as “mulit-agent systems”.

Correctness: Yes.

Clarity: This paper is well-written and readable.

Relation to Prior Work: See Weakness W4.

Reproducibility: Yes

Additional Feedback: My main concern is the limited evaluation and scalability issue of the proposed approach, which could be addressed either through theoretical analysis or empirical validations. However, the current version does not provide this, which should be addressed before ready for publication in top revenues.


Review 3

Summary and Contributions: - The authors set out to investigate networked information structures for multi-agent RL in common-pool resource management tasks. In particular, their goal is to investigate "networked system inefficiencies" in multi-agent RL systems applied to CPR management problems. - Networked system inefficiencies (NSI) are operating conditions of a learned networked system under which all the potential equilibria identified to exist within the system are inefficient. The NSI conditions are equivalent to the conditions in the definition of an intertemporal (sequential) social dilemma given in Hughes et al. (2018). This is the right thing to do since the goal here is to carry out a thorough analysis of a new problem along the lines recommended by that line of previous work, and simultaneously to extend these ideas to networked games for the first time. - The authors find that differentiable communication protocols lead to more cooperation than non differentiable protocols. Their Schelling diagram games have equilibria closer to average welfare maximizing points. Also, an algorithm called NeurComm performs particularly well in this domain.

Strengths: - The authors correctly apply the empirical game theoretic approach of Leibo et al. (2017), Perolat et al. (2017), and Hughes et al. (2018) to a new problem involving a network structured game. - Plotting a separate Schelling diagram for each algorithm, as in figure 4, is a very nice way to think about the various options for adaptive institutions in this system. It's quite interesting to see how the shape of the Schelling diagram varies so much with the choice of algorithm. - I appreciated the argument about safety critical applications of MARL, why they entail serious engagement with the topic of this paper, and that the specific example of water use management would be one such application (if scaled up and applied in the real world).

Weaknesses: - In multi-agent reinforcement learning research, Schelling diagrams are normally plotted as a function of the number of *other cooperators* (besides the focal agent making the decision), i.e. |C| - 1, rather than the total number of cooperators, |C|, as was done here. Either way is certainly correct in principle, Schelling said as much in the original 1973 paper. However, there are several reasons why the |C| - 1 parameterization is convenient. For instance, it lets you read off game theoretic properties from the diagram more easily. To see if cooperation or defection is favored for a particular number of other cooperators, you simply compare a point on the R_c curve to the point on the R_d curve that is right above it. This is because the |C|-1 parameterization shifts the R_c and R_d curves relative to one another. Using the authors' parameterization by |C| instead, you have to compare diagonally across the diagram: between R_c(|C|) and R_d(|C| - 1). Being able to compare vertically between the curves in the |C| - 1 parameterization makes all the game theoretic properties that can be discerned from the diagram easier to see. For example, it is immediately obvious when either strategy is dominant and the locations of Nash equilibria are immediately obvious. It also simplifies the statement of condition C3 on page 7 of the paper. - Why are there no numbers labeled in the NSI indicator inset in Fig. 3D (and 4B-H)? It's clear from the Schelling diagram what some of their values should be of course, but it couldn't hurt to label them as well. if only to help the reader check their understanding. - How the third bar in the NSI indicator was computed is not clear (insets in all Schelling diagram plots). The text says that it pertains to condition C3, but that condition says "for all n \le i and/or n \ge j". So which number is the value shown in the bar of the NSI indicator? Is it the minimum R_d(n-1) - R_c(n) for any choice of n in that range? - What is the precise meaning of the shaded ovals drawn on the Schelling diagrams? The paper says they indicate Nash equilibria. But strictly speaking, a Nash equilibrium should be a strategy profile, not a set of payoffs like is shown on these diagrams. Something that could be indicated by highlighting points on a Schelling diagram is a summary of the payoffs obtained by all agents when they play a strategy profile in Nash equilibrium. This is presumably what the authors meant to say they were indicating with the description of the filled ovals in lines 260 - 261. - Most of Fig. 4A looks immediately sensible to me, however it would have been nice to explain why the first three algorithms show a decrease in restraint between the second to last column (rate = 0.042) and the last column (rate = 0.03). I think it's clear from context what must have happened here. Defection becomes favored in these very scarce conditions because conditions are so bad that cooperation is not profitable. But it would be nice if the authors had explained this, again, if only for the reader to be able to check their understanding is correct. - It is not clear why NeurComm performs the best. Is there anything about how this algorithm works that we should consider in order to understand why this result was the way it was? Why do the C and D strategies seem to get approximately equal payoffs for NeurComm but not for any of the other algorithms? - I know this is covered to some extent in the cited references, but I still think a paragraph should have been devoted to the topic of the relationship between the empirical game represented by the Schelling diagram and the original Markov game. This is an active area with several unknowns that bear directly on the topic of this paper. For example, at very least the relationship between equilibria of the empirical game and the underlying Markov game should be discussed in the paper. - There should be more discussion of the relationship between Networked System Inefficiencies (NSIs) and social dilemmas. Since we already have the well established concept of a social dilemma, why do we need to also define this new concept of NSI? Is there something new that it brings to the table? - The conclusion that the the communicating algorithms achieve equilibria that are close to optimal in the sense of average reward, but still deficient from the perspective of the NSI criteria bears some extra examination. It should be pointed out that this result appears from the Schelling diagrams to be a consequence of the small number of players in the game. For example, looking at the Schelling diagram for the DIAL algorithm, if you were to keep the exact same shape of the Schelling diagram, but increase the number of players in the game to create more possible locations on the X-axis between the points currently labeled n = 2 and n = 3, then the equilibrium would correspondingly move to the left. But the point where the average payoff is maximal would remain in the same place. The same player number inflation argument works with the Schelling diagrams for any of the communicating algorithms. I think this argument only slightly weakens the conclusion about the communicating algorithms being more effective than the non-communicating algorithms, after all it assumes (unrealistically) that the Schelling diagram would not change shape if the number of players increased. Moreover, this has no bearing whatsoever on the result for the NeurComm algorithm. Unlike all the others, the Schelling diagram for NeurComm is such that the Nash equilibrium would not shift to the left even if we dilated the number of players. - OK, I've got one more comment on this part. The author's stated goal in introducing NSIs is to study the operating conditions of a learned networked system under which *all* the potential equilibria identified are inefficient. It's not clear that the formal definition offered for NSI fully meets this objective. The key word is *all*. It looks like the NSI conditions pick out Schelling diagrams for which *there exist* at least one inefficient equilibrium. But I don't think they need imply *all* equilibria of the game represented by the Schelling diagram are inefficient. Of course, the authors could easily fix this by changing the quantification the statement of their goal (lines 235 - 237). - Please note that I would gladly increase my rating for this paper if the authors are able to satisfactorily address the points I've raised here.

Correctness: Yes

Clarity: Yes, I very much enjoyed reading it.

Relation to Prior Work: The relation to prior work is very well described throughout the paper. It's very clear and well-written. I was convinced by the explanations provided for how this work differs from prior work in this area. The synthesis pursued here of networked MARL ideas with the line of research pursuing EGTA+MARL+CPRs is certainly a sufficiently large and novel contribution beyond previous work.

Reproducibility: Yes

Additional Feedback: Minor comments: - Unlike how "cooperative" is the correct adjective for an agent that cooperates, the correct adjective for an agent that defects is "defecting" not "defective". Saying the agent is defective actually means (incorrectly) that the agent is broken. - In the caption of Fig. 3D (the Schelling diagram), it would be helpful to either add the explanation of what the NSI indicator is, and what the filled oval indicates, or alternatively, the caption could just mention these features of the plot and say "see text" instead of repeating their explanation in both places. - It's a bit annoying that the algorithms are not listed in the same order between Fig. 4A and Figs. 4B- 4H (Looks like NeurComm was moved from the bottom of the list up to the more prominent position at the top, next to IA2C. Why not just reorder them in Fig. 4A to match the order you want to present them in the rest of the subfigures? - While I certainly appreciate the sentiment being expressed in line 316-317 where the authors began a sentence with the phrase "It is our belief that...", it is not a strong way of arguing in academic writing to refer to one's own belief as any kind of reasoning or support for any claim. The paragraph could easily be made more persuasive by removing the subjective language.


Review 4

Summary and Contributions: The manuscript describes a multi-agent reinforcement learning approach for monitoring control tasks. The authors also perform an experimental, game-theoretic analysis on managing a water treatment system.

Strengths: The insights and comments, modeling a real problem (water management) as a multi-agent control problem, and identifying the particular challenges in the actual problem.

Weaknesses: Limited technical methodological contributions: both, no new approaches and no novel analysis methods.

Correctness: Yes, they are correct.

Clarity: Well written paper.

Relation to Prior Work: Very little prior work was done in this area, to my knowledge.

Reproducibility: Yes

Additional Feedback:

[Author Response · NeurIPS 2020]

We thank all the reviewers for their comments on our work as well as for their suggestions on how it can be improved.
We especially want to thank reviewer 3 for their detailed and insightful review, and for highlighting specific and
important shortcomings of our original submission. We hope that the following will address all major concerns raised.

**Additional analysis on NeurComm** (R1, R2, R3): Almost all the reviewers requested that we look at the data more
closely and attempt to understand why NeurComm performed the way it did. Although the primary goal of the EGTA
was to highlight the differences in solution concepts between algorithms, we agree that understanding the reason why
these differences exist is also important and will improve the quality of our paper. In this spirit, we have taken the
advice of R2, and computed the metrics from [1], namely, the Utilitarian, Equality and Sustainability metric for each
algorithm for the two cases where (1) all agents cooperate and (2) agents play their equilibrium strategy, shown in Fig.
1A-C. When all agents are cooperating, the Equality for NeurComm is lower (i.e. the distribution of reward among
agents is less uniform) than that of the other communicating algorithms, however, it is still able to achieve both a higher
score for the group (more utilitarian) and have higher levels of sustainability. This provides some supporting evidence
to our claim that NeurComm is perhaps better able to learn how to coordinate agents effectively. We have included this
analysis in our paper, with additional discussions and interpretations of the results.

**Connections to prior game-theory work** (R1, R3): We thank R1 for pointing us towards the valuable literature
on network games (e.g. [2]). We will include a thorough discussion of this line of work in the context of our own.
Furthermore, we thank R3 for suggesting that we connect our analysis to the underlying Markov game. Specifically,
we used the finite sample analysis in [3] combined with bootstrap resampling (please see Fig. 1D for an example), to
improve our estimates, obtain confidence intervals and ensure a large enough sample size to tightly bound the difference
between our empirical estimates and the original game. We have included this analysis in our work.

**Introduction of NSIs** (R1, R3): The terms used in the literature on SSDs have a strong association with analysing
human-like agents. For example, two of the conditions characterising a social dilemma are often referred to, respectively,
as "fear" and "greed" [4], which in the context of an engineering system would seem an anthropomorphisation of the
incentive to defect. Our aim with introducing the NSI was to remove this additional layer of association. Therefore,
we combined the fear and greed conditions to form $C_3$ in our paper and opted for terminology such as "inefficiency"
instead of "social dilemma". We have taken care to add the above motivation to the paper and improve the clarity
behind the term. However, we are aware that social dilemmas are long established concepts in game theory and if a
final suggestion is given that we remain with the SSD terminology to make the paper more accessible, we will do so.

**R3** (*Schelling plots and meaning of ovals; NSI definition and computation*): We have changed our Schelling diagrams to
use the $|C| - 1$ parameterisation to improve readability and clarified the meaning of the shaded ovals. As R3 points out,
these are meant to refer to the payoffs obtained by agents playing a strategy profile in Nash equilibrium. Furthermore,
to compute the value in the NSI indicator related to $C_3$, we used the value that is the maximum of $R_d(n - 1) - R_c(n)$
over the range specified in the paper. We have noted the calculation explicitly in the paper for the new parameterisation.
We have also updated our stated goal surrounding the identification of NSIs. We agree with R3's suggestion that it
makes more sense to refer to NSIs as situations where there *exists* at least one inefficient equilibrium. For example, in
our original definition, IA2C would not have strictly been classified as an NSI, which is inconsistent with the indicator.
Finally, we have added a more detailed explanation regarding the last two columns of Fig. 4A (also requested by R2),
which is in line with R3's interpretation.

**Misc**: We have made the following corrections/changes as suggested: correctly labelling the equilibrium in Fig. 4D
(R1); correcting all spelling, enlarging Fig. 4 for improved readability (R2); using the correct term for defecting agents,
adding an explanation of the NSI indicator in Fig. 3D and numerical values, changing the order of the plots in Fig.
4B-H to match the order in Fig. 4A and removing the subjective language in the conclusion (R3). We greatly thank the
reviewers for improving our work with these suggestions.

Figure 1: *Social metric and finite sample analysis (best viewed zoomed in).* (**A**) Utilitarian (**B**) Equality (**C**) Sustainability (**D**) Bootstrap estimation.

[1] J. Perolat, J. Z. Leibo, V. Zambaldi, C. Beattie, K. Tuyls, and T. Graepel, "A MARL model of common-pool resource appropriation," NeurIPS, 2017.
[2] Jackson, Matthew O., and Yves Zenou. "Games on networks." In Handbook of game theory with economic applications, 2015.
[3] Tuyls, Karl, Julien Perolat, Marc Lanctot, Joel Z. Leibo, and Thore Graepel. "A generalised method for empirical game theoretic analysis." AAMAS, 2018.
[4] Hughes, Edward, Joel Z. Leibo, Matthew Phillips, Karl Tuyls, Edgar Dueñez-Guzman, Antonio García Castañeda, Iain Dunning et al. "Inequity aversion improves
cooperation in intertemporal social dilemmas." NeurIPS, 2018.


[Meta-Review · NeurIPS 2020]

The paper is modelling MARL problems under the angle of social dilemma, and tries to tackle the problem of common-pool resource management. The authors do not introduce a novel method, instead this paper is a comparison of a wide range of existing relevant algorithms on a single problem (water management). The experiments are well motivated and in general, the paper is very clear. My understanding is that although the paper focuses on a water management, it is aimed as a more general survey of the quality of current MARL algorithms on common-pool resource management. The authors argue that water management is a good example to study because it is critical and life-supporting, and safety issues are very relevant. One weakness is that only one problem is studied, although the paper is presented as studying a problem that is more general than water management. The problem is also small scale, although reviewers disagree on whether this is too small to be meaningful. Reviewers also point out that some baselines are missing. Reviewers also pointed out that the paper does not attempt to analyse the results (especially on NeurComm, which seems to be better at finding a stable equilibrium), but the authors have addressed it in their author feedback and have included more analysis to that end. Meta-reviewers suggest that the authors should flesh out their broader impact statement to consider what the implications would be if a deployed MARL algorithm were to fail.